# The Relationship between Stretching Intensity and Changes in Passive Properties of Gastrocnemius Muscle-Tendon Unit after Static Stretching

**DOI:** 10.3390/sports8110140

**Published:** 2020-10-23

**Authors:** Taizan Fukaya, Masatoshi Nakamura, Shigeru Sato, Ryosuke Kiyono, Kaoru Yahata, Kazuki Inaba, Satoru Nishishita, Hideaki Onishi

**Affiliations:** 1Institute for Human Movement and Medical Sciences, Niigata University of Health and Welfare, 1398 Shimami-cho, Kita-ku, Niigata City, Niigata 950-3198, Japan; masatoshi-nakamura@nuhw.ac.jp (M.N.); hpm19006@nuhw.ac.jp (S.S.); hpm19005@nuhw.ac.jp (R.K.); hpm20011@nuhw.ac.jp (K.Y.); onishi@nuhw.ac.jp (H.O.); 2Department of Rehabilitation, Kyoto Kujo Hospital, 10 Karahashirajoumoncho, Minami-ku, Kyoto 601-8453, Japan; 3Department of Physical Therapy, Niigata University of Health and Welfare, 1398 Shimami-cho, Kita-ku, Niigata City, Niigata 950-3198, Japan; hpa16021@nuhw.ac.jp; 4Graduate School of Medicine, Kyoto University, 53 Kawara-cho, Shogoin, Sakyo-ku, Kyoto 606-8507, Japan; satoru@rehalab.jpn.org; 5Institute of Rehabilitation Science, Tokuyukai Medical Corporation, 3-11-1 Sakuranocho, Toyonaka, Osaka 560-0054, Japan; 6Kansai Rehabilitation Hospital, Tokuyukai Medical Corporation, 3-11-1 Sakuranocho, Toyonaka, Osaka 560-0054, Japan

**Keywords:** static stretching, range of motion, passive torque, stretching intensity

## Abstract

This study aimed to investigate the relationship between relative or absolute intensity and changes in range of motion and passive stiffness after static stretching. A total of 65 healthy young adults voluntarily participated in this study and performed static stretching of the plantar flexor-muscle for 120 s. Dorsiflexion range of motion and passive torque during passive dorsiflexion before and after stretching were assessed. We measured the passive torque at a given angle when the minimum angle was recorded before and after stretching. The angle during stretching was defined as the absolute intensity. Dorsiflexion range of motion before stretching was defined as 100%, and the ratio (%) of the angle during stretching was defined as the relative intensity. A significant correlation was found between absolute intensity and change in passive torque at a given angle (r = −0.342), but relative intensity and range of motion (r = 0.444) and passive torque at dorsiflexion range of motion (r = 0.259). A higher absolute intensity of stretching might be effective in changing the passive properties of the muscle-tendon unit. In contrast, a higher relative intensity might be effective in changing the range of motion, which could be contributed by stretch tolerance.

## 1. Introduction

Flexibility has been defined as the ability to move a joint with its complete range of motion (ROM). ROM plays an important role in sports performance and the ability to carry out activities of daily living [1,2], which might influence muscle strain injury risk [3]. Stretching is performed in several fields to increase ROM. Static stretching (SS) is one of various stretching methods. A systematic review has reported that ROM increases after SS [4]. Moreover, this review suggests that mechanisms of ROM increment could be due to a change in the sensation to tolerate loading before terminating the stretch (i.e., stretch tolerance) and change in viscoelastic properties of the muscle-tendon unit [4].

Recently, Apostolopoulos et al. (2015) have reported that the four stretch parameters associated with an increase in flexibility after SS are (1) intensity, (2) duration, (3) position, and (4) frequency [5]. In particular, previous studies have reported that a longer stretching duration is more effective in changing ROM and the viscoelastic properties [6,7]. Additionally, a higher stretching intensity has been found to be more effective in inducing a change in ROM and stretch tolerance, as performed by the same subjects at a different intensity of SS [8]. Similarly, Kataura et al. (2017) have reported that a high-intensity SS has a greater effect on changes in ROM and passive stiffness than low-intensity SS [9]. Conversely, Santos et al. (2019) have reported that ROM significantly increases after SS, but no significant difference was observed in ROM change between high- and low-intensity SS [10]. Therefore, no consensus was found on the effect of stretching intensity on ROM and muscle-tendon stiffness.

As described above, the method of setting stretching intensity might be related to why the consensus on the effect of stretching intensity on ROM and passive property was not achieved. A previous study has reported that 120% ROM stretching intensity affects ROM and the passive properties compared to 100% ROM stretching intensity for the same subjects, i.e., relative intensity. Conversely, a previous study has reported that there is no significant difference in ROM change between the stretching intensity defined as the angle when high or low sensation of stretching felt for each subjects as the stretching intensity [10]. Therefore, it is possible that the difference in setting stretching intensity could influence the effects of SS on ROM and passive property. Overall, we assumed that the changes in ROM and passive properties might be affected by relative intensity when compared between same subjects. In most previous studies, SS was performed at the onset of the feeling of pain or discomfort, which was defined as the SS intensity. That is, SS was performed at different absolute angles between each subject. Therefore, it is assumed that the changes in ROM and passive properties might be influenced by the stretching angle, that is, absolute intensity, when compared among individuals although the relative intensity was the same between subjects. However, to the best of our knowledge, no study has investigated the effect of SS with the difference of relative or absolute intensity on ROM and passive properties.

This study aimed to investigate the relationship between relative or absolute intensity and the effect of SS on ROM and passive properties. Previous studies have reported that higher-intensity SS affects ROM changes and muscle-tendon stiffness when compared within same subjects [9,11]. On the other hand, as described above, no significant difference was observed between ROM changes and stretching intensity when compared between each subject [10]. Therefore, it was hypothesized that a higher relative intensity of SS is more effective in inducing changes in ROM and passive properties, but absolute intensity of SS might not be related to changes in ROM and passive properties.

## 2. Materials and Methods

### 2.1. Participants

A total of 65 healthy young adults (male, 37 students; females, 28 students) voluntarily participated in this study (age, 21.0 ± 0.2 years; height, 166.7 ± 8.5 cm; body mass, 59.8 ± 10.0 kg). Participants who had a history of an operation performed on their back or lower extremity, lower-extremity contracture, neurological disorders, or took hormone or muscle-affecting drugs were excluded. Written informed consent was obtained from all the participants. This study was approved by the ethics committee of our institution (17677).

G*Power software (v 3.0.10; Dr. Franz Faul, Kiel University, Kiel, Germany) was used to calculate the sample size on the effect size for correlation analysis (effect size = 0.3 (medium), α = 0.05 and power = 0.80), and elicited result have suggested that the minimum required subjects’ number was 64 for this study.

### 2.2. Study Design and Overview

A pre- and post-stretching evaluation was conducted. The participants performed 120 s of SS using the plantar flexor muscles [7]. Dorsiflexion (DF) ROM, passive torque at DF ROM [12], and passive torque at a given angle [13] were measured before (PRE) and immediately after SS (POST). Additionally, the angle during SS was measured as stretching intensity, and the relationship between the stretching intensities and DF ROM or passive torque at a given angle was investigated (Figure 1).

### 2.3. Procedures

The participants were seated in an isokinetic dynamometer (Biodex system 3.0; Shirley, NY, USA) chair at 0° knee angle (i.e., the anatomical position) and 70° hip flexion to prevent tension at the back of the knee, with adjustable belts over the trunk and pelvis and the ankle fixed to the footplate [14]. Then, the participants passively moved the footplate of the dynamometer starting from the ankle at neutral angle (0°) to the dorsiflexion angle at the point of feeling pain or discomfort at 5°/s speed, and the torque-angle curve was measured [15].

DF ROM, passive torque at DF ROM, and passive torque at 100% ROM were measured by calculating the torque-angle curve using an isokinetic dynamometer [6] (Figure 2). DF ROM (°) was defined as the dorsiflexion angle at the point of feeling pain or discomfort. Passive torque at DF ROM (Nm) was defined as the passive torque at the point of maximum dorsiflexion [14]. Passive torque at a given angle (Nm) was defined as the passive torque at the point when DF ROM was the least before or after SS [13]. The passive torque at a given angle was the index that indicted the passive resistance of the muscle–tendon unit [16], and the comparison of passive torque at a given angle indicated a change in the passive stiffness of the muscle-tendon unit.

### 2.4. Static Stretching

SS was performed with the plantar flexor muscles in a similar position for the measurement. The participants passively moved the footplate of the dynamometer from the ankle at neutral angle to the maximum dorsiflexion angle without feeling pain [17] at 5°/s speed and held the position for 120 s at the angle [7,18].

### 2.5. Definition of Stretching Intensity

We defined absolute intensity and relative intensity as the stretching intensity. It was defined that absolute intensity reflected the intensity of stretching among subjects. On the other hand, it was defined that relative intensity reflected the intensity of stretching within each subject. Using the dynamometer, we measured the DF angle during SS in a similar position for the static stretching. The participants passively moved the footplate of the dynamometer from the ankle at a neutral angle to the DF angle at the point of feeling of pain or discomfort while their ankles were dorsiflexed. During SS, the DF angle (°) was defined as the absolute intensity. On the other hand, DF ROM at PRE which we measured before stretching was defined as 100% [19], and during SS, the DF angle ratio (%) was defined as the relative intensity as following formular:(1)$$\mathrm{Relative}\;\mathrm{intensity}\;\left(\%\right)=\;\frac{DF\;angle\;during\;SS}{DF\;ROM\;at\;PRE}$$

### 2.6. Measurement Reliability

Before this study, two tests were performed on eight healthy young adults on different days to determine the test-retest reliabilities for DF ROM, passive torque at DF ROM, and passive torque at a given angle. The calculated intraclass correlation coefficients for DF ROM, passive torque at DF ROM, and passive torque at a given angle were 0.94, 0.91, and 0.89, respectively, indicating that the reliability was high for all outcome measures [20].

### 2.7. Statistical Analyses

Paired *t*-test was used to compare the differences in DF ROM between PRE and POST, passive torque at DF ROM, and passive torque at a given angle. Using the t-value and sample size, the effect size (r) was calculated. The effect size classification was set where r < 0.1 was considered trivial, 0.1–0.3 was considered small, 0.3–0.5 was considered moderate, and >0.5 was considered large. Moreover, using the standard error of the mean, we calculated the minimal detectable change (MDC). The relationship between absolute or relative intensities and changes in DF ROM, passive torque at DF ROM, and passive torque at a given angle was determined using Pearson’s product–moment correlation coefficient. All statistical analyses were performed using R2.8.1 (CRAN, freeware, version 2.8.1, R Foundation for Statistical Computing, Vienna, Austria), and significance was set at *p* < 0.05. All the data are presented as mean ± standard deviation.

## 3. Results

### 3.1. Effect of Static Stretching on Dorsiflexion Range of Motion, Passive Torque at Dorsiflexion Range of Motion, and Passive Torque at a Given Angle

DF ROM significantly increased after SS (PRE: 26.6° ± 9.5°, POST: 29.6° ± 9.4°, *p* < 0.01, r: 0.59, MDC: 4.2°). However, no significant change was observed in passive torque at DF ROM (PRE: 37.4 ± 14.4 Nm, POST: 37.6 ± 14.2 Nm, *p* = 0.83, r: 0.03, MDC: 6.2 Nm). Passive torque at a given angle significantly decreased after SS (PRE: 33.8 ± 13.0 Nm, POST: 29.7 ± 10.9 Nm, *p* < 0.01, r: 0.68, MDC: 7.4 Nm).

### 3.2. Relationship between Stretching Intensities and the Changes in Passive Property of Muscle-Tendon Unit

The absolute intensity and relative intensities of SS were 24.2° ± 8.9° and 91.6% ± 11.6%, respectively. Figure 3 and Figure 4 show the relationships between these stretching intensities and DF ROM changes, passive torque at DF ROM, and passive torque at a given angle, respectively. No significant correlations were observed between the absolute intensity and DF ROM changes (r = −0.029, *p* = 0.82, Figure 3A) and passive torque at DF ROM (r = −0.195, *p* = 0.119, Figure 3B). However, a significant negative correlation was found between the absolute intensity and change in the passive torque at a given angle (r = −0.342, *p* = 0.005, Figure 3C).

Additionally, Pearson correlation coefficients showed positive correlations between the relative intensity and changes in DF ROM (r = 0.444, *p* = 0.0002, Figure 4a) and passive torque at DF ROM (r = 0.259, *p* = 0.037, Figure 4B). However, no significant correlation was observed between the relative intensity and passive torque at a given angle (r = −0.231, *p* = 0.065, Figure 4C).

## 4. Discussion

This study investigated the relationships between the absolute or relative intensities and DF ROM, passive torque at DF ROM, and passive torque at a given angle after a 120 s SS. The results of the study showed a significant negative correlation between the absolute intensity and passive torque at a given angle (Figure 3C). In addition, this study showed significant positive correlations between the relative intensity and changes in DF ROM and passive torque at DF ROM (Figure 4A,B). To the best of our knowledge, this study is the first to describe a different relationship between absolute intensity or relative intensity on the changes in ROM and passive properties after SS.

This study showed that the DF ROM significantly increased after a 120 s SS. Generally, the mechanism of the increase in DF ROM is related to the increase in the ability to tolerate loading prior to stretch termination (the increase in stretch tolerance) [12,21] and the change in viscoelastic properties (e.g., the reduction in passive torque at a given angle) [7]. Furthermore, this study showed that the DF ROM significantly increased and passive torque at a given angle significantly decreased after 120 s SS, indicating that results were similar to that of a previous study [7]. However, no significant difference was observed in the passive torque at DF ROM after SS. Kay et al., have reported that no significant change is observed in the passive torque at DF ROM, whereas a significant positive correlation is observed between increased DF ROM and passive torque at DF ROM. They concluded that an increased stretch tolerance contributes to an increase in ROM [22]. In this study, a significant correlation was observed between changes in DF ROM and passive torque at DF ROM (r = 0.77, *p* < 0.05, data not shown). Therefore, our results suggest that the DF ROM increase might have resulted from an increase in the stretch tolerance and a change in the passive stiffness of the muscle-tendon unit.

Regarding the relationship between absolute intensity, i.e., DF angle of SS, and changes after SS, our results showed that the absolute intensity was significantly negatively correlated with the change in passive torque at a given angle, suggesting that the higher absolute intensity greatly decreases the passive torque at a given ROM when compared among individuals. The passive torque at a given angle has been shown to indicate the resistance of the muscle-tendon unit [16], and it has been suggested that a decrease in the passive torque at a given angle after SS results in a change in the passive stiffness of the muscle-tendon unit. We defined the angle during SS as the absolute intensity among different subjects, and our results were similar to those of a previous study that investigated the intensity of SS for the same subjects [9]. Therefore, our results suggest that the higher absolute intensity, i.e., larger angle of SS, might contribute to a greater mechanical stress in the muscle-tendon unit and might result in a change in the passive stiffness of the muscle-tendon unit. Conversely, no significant correlation was observed between the absolute intensity and DF ROM or passive torque at DF ROM. In this study, SS was performed at the maximum DF angle by each subject without feeling pain, and this angle was defined as the absolute intensity. Therefore, the absolute intensity varied among the subjects, but no significant correlation was observed between the absolute intensity and change in passive torque at DF ROM (stretch tolerance). Stretch tolerance was defined the capacity to tolerate loading prior to stretch termination [4]. Thus, it is suggested that stretch tolerance was not influenced on absolute intensity because absolute intensity was similar to the sensation felt by the subjects prior to the pain. Therefore, this study found no significant correlation between the DF ROM changes and absolute intensity since no significant correlation was observed between stretch tolerance and absolute intensity.

On the other hand, a significant positive correlation was observed between the relative intensity, i.e., the angle during SS per DF ROM at PRE (%), and changes in DF ROM and passive torque at DF ROM. These results were consistent with those of the previous study that investigated the relative intensity among same subjects [21]. A previous study has shown that an increase in passive torque at DF ROM leads to a change in the stretch tolerance [12]. Therefore, our result showed that the higher SS intensity performed by each subject without feeling pain was more effective in increasing the stretch tolerance, which could have contributed to the significant change in DF ROM. Interestingly, the study results showed that there was no significant correlation between relative intensity and change in the passive torque at a given angle, suggesting that the change in the passive stiffness of the muscle-tendon unit could contribute to the absolute intensity but not relative intensity. In other words, our results suggest that a greater angle of SS (a higher absolute intensity), regardless of each subject’s intensity, might play an important role in changing the passive stiffness of the muscle-tendon unit.

In this study, the difference in results were found between absolute intensity and relative intensity. There was a significant correlation between absolute intensity and the changes in passive torque at a given angle. Morse et al. (2018) reported that, with increasing dorsiflexion angle, both muscle and tendon elongation increased linearly [15]. Besides, Nakamura et al., declared that there was a significant correlation between the rate of change in muscle elongation and that in passive torque during SS [7]. Therefore, there is a possibility that a greater angle of SS, that is, higher absolute intensity, could cause a greater muscle elongation and result in a greater change in passive torque at a given angle (i.e., passive stiffness of muscle-tendon unit). On the other hand, there was a significant correlation between relative intensity and the change of DF ROM and passive torque at DF ROM (i.e., stretch tolerance). In a previous study, it was reported that an increased ROM contributes to an increase in stretch tolerance [22]. Moreover, previous studies have suggested that a reduction in the perceptions of pain and discomfort, accompanied by a change in neural and psychological factors after stretching cause the increase in stretch tolerance [23,24]. Possibly, the relative intensity is depending on each subject’s sensation because the intensity was determined from the angle when each subject felt pain or discomfort. Therefore, it is assumed that a tolerable greater angle between individuals has a greater effect on the changes in stretch tolerance, which resulted in an increase in ROM.

In previous studies, it has been found that a poor ROM [25,26] and increased muscle stiffness [27,28] might lead to sports injury. Therefore, the increase in ROM or the decrease in passive stiffness could be important to decrease the risk of sports injury. Considering the results of this study, when there was an impairment ROM or passive stiffness, we should choose the absolute or relative intensity. Thus, to increase ROM, we could perform SS of higher relative intensity. On the other hand, to decrease passive stiffness, we could perform SS of higher absolute intensity.

This study had three main limitations. First, the effect of SS on flexibility may have differed depending on the sex. Miyamoto et al., have suggested that muscular factors associated with ROM are different between men and women [29]. In this study, no significant differences in changes in passive properties (ROM and passive torque) were observed between male and female subjects (data not shown). Moreover, few studies have investigated the difference in the effect of SS between men and women; furthermore, previous studies have not investigated the effects of SS separately for men and women [22,30]. Second, healthy young adults were enrolled in this study. For the healthy people, higher-intensity SS caused pain during SS; however, the pain disappeared immediately after SS [31]. However, a previous study has suggested that a higher-intensity SS may cause an inflammatory response [5]. Stretching is performed by patients with various illnesses, and the effects of SS may vary among individuals. Therefore, the effect of SS intensity on ROM and passive properties should be investigated in different study cohorts. Third, this study investigated the relationship between stretching intensity and DF ROM and passive properties immediately after SS. However, a previous paper suggested that, with a period of 3–8 weeks, the chronic SS did not seem to change either the muscle or the tendon properties [32]. Thus, since this study results might not be applicable to the SS chronic effects, further studies are needed to investigate this relationship after chronic SS program.

## 5. Conclusions

This study investigated the relationship between absolute and relative intensities of SS and the changes in DF ROM, passive torque at DF ROM, and passive torque at a given ROM after a 120 s SS. The study results showed that there was a significant negative correlation between the absolute intensity and passive torque at a given ROM. Absolute intensity was calculated from the angle of SS and was not different among subjects. Therefore, we suggest that a greater angle during SS is more effective in creating change in the passive stiffness of the muscle-tendon unit. On the other hand, there was a significant positive correlation between the relative intensity and ROM and passive torque at DF ROM. Relative intensity was calculated from the angle during SS per DF ROM at PRE (%) and was influenced by the subject’s sensation. Thus, in an individual, SS of a tolerable maximum angle was more effective in inducing a change in the stretch tolerance, which resulted in an increase in DF ROM.

## Figures and Tables

**Figure 1 sports-08-00140-f001:**
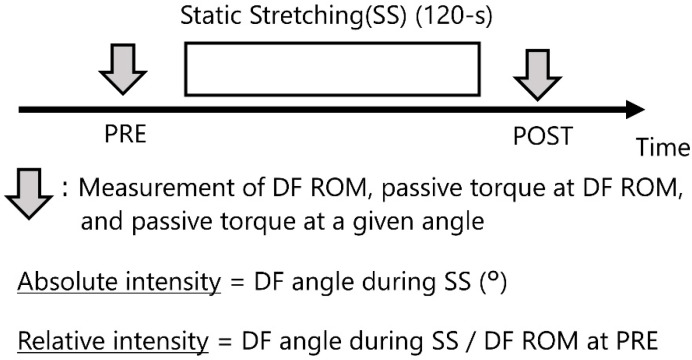
Timeline of the static stretching (SS) protocols. Dorsiflexion (DF) range of motion (ROM), passive torque at DF ROM, and passive torque at a given angle were measured before (PRE) and after (POST) SS. SS was performed for 120 s. DF ROM during SS was defined as absolute intensity. The ratio of DF ROM at PRE to DF angle during SS was defined as relative intensity.

**Figure 2 sports-08-00140-f002:**
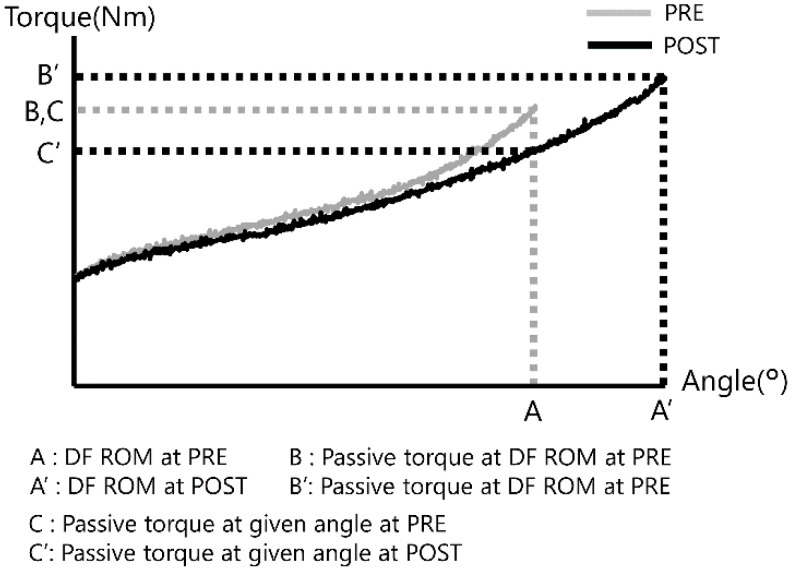
Typical passive torque-angle curves of a patient at pre- and post-stretching. Passive torque at a given angle was determined as the passive torque at the point DF ROM was the least before and after static stretching.

**Figure 3 sports-08-00140-f003:**
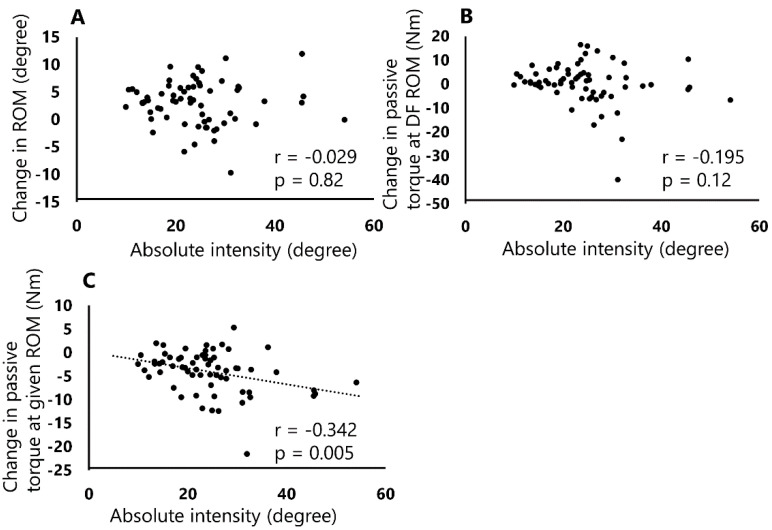
Correlations between absolute intensity and changes in DF ROM (**A**), passive torque at DF ROM (**B**), and passive torque at a given angle (**C**). Significant correlation was observed between absolute intensity and change in passive torque at a given angle (**C**; r = −0.342, *p* = 0.005).

**Figure 4 sports-08-00140-f004:**
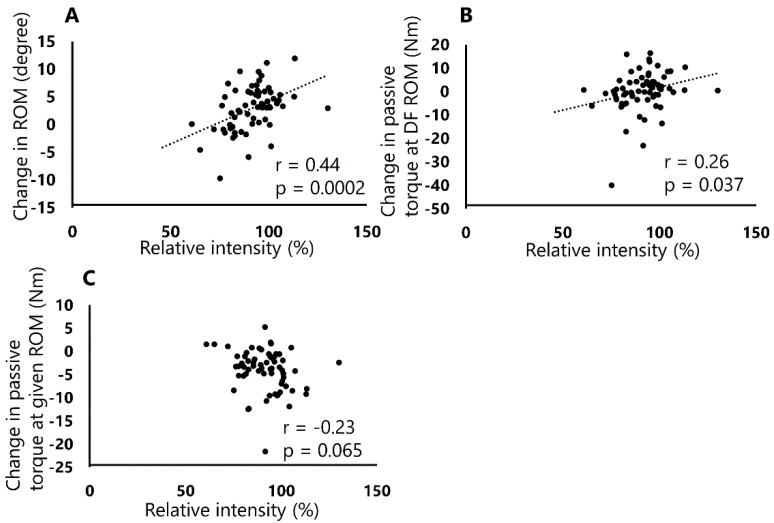
Correlations between relative intensity and DF ROM (**A**), passive torque at DF ROM (**B**), and passive torque at a given angle (**C**). Significant correlations were observed between relative intensity and changes in DF ROM (r = 0.444, *p* = 0.0002) and passive torque at DF ROM (r = 0.259, *p* = 0.037).

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
