# Peer review of "The Relationship between Stretching Intensity and Changes in Passive Properties of Gastrocnemius Muscle-Tendon Unit after Static Stretching"

_sports, 2020, doi:10.3390/sports8110140_

Round 1

Reviewer 1 Report

Thank you for submission. Overall, the manuscript is well-written but I have several areas that should be improved prior to publication.

References: You cite a few papers by Freitas et al. but not their 2018 Review. You may want to consider reviewing that as well, if you haven't already.

Intro

Can you better articulate (Line 67-68) why relative intensity is better? You provide several references to support the study, but I do not believe you clearly articulate the why relative intensity might be better (I note this in the discussion as well). As written the paper is very dry but lacks some readability for those unfamiliar with the details of this line of research.

Line 72 - I would split the Purposes and hypotheses into a final paragraph, which would reduce the length of the current paragraph and more clearly delineated the purposes while reading.

Methods

Line 115 - Can you expound a bit more on stretching intensity? It reads a bit vague for those less familiar with this area.

In particular, perhaps better describe relative intensity.

Statistical Analyses

Please provide details on effect sizes for differences and relate to clinically meaningful changes. It is one thing to say that there's a statistical differences in degrees, but are there applicable real-world differences.

Results

Please be sure to report effect sizes with full p-values rather than p < 0.0X. Can you improve your figures. They're very small and difficult read.

Discussion

General comments: As I see it, the figures for absolute are opposite of those for relative. Can you better articulate the relevance of this in the discussion?

In addition, can you better articulate the relevance and applicability of your findings to reach a larger audience?

One significant problem with your paper actually relates to your limitations. You state that SS may differ between male and female subjects and little has been studied in this area, but you could actually report on this! You have 37 men and 28 women, why not run an analysis looking at whether the distributions are normal and compare? IMO, this should be a major part of your paper here. This would increase the significance of the paper and interest in more readers. I recommend you add this and drop that limitation. With the above stated, you may want to rethink your discussion.

Conclusions

Try to make this more reader friendly with some less technical language to allow some readers to better see applicability of the research.

Reviewer 2 Report

The study is focused on a interesting topic, it is overall well written and could add to the general level of knowledge of this argument. 

However, in my opinion, it is in the present form a little poor in terms of novelty and outcomes and could need some deeper analyses.

It is interesting the approach used here, and the use of a isokinetic apparatus makes the outcomes reliable and easily analyzable. This said, it would be really interesting and would strongly add to the study to have some information about the changes in the passive torque during the maintenance of the relative angle. The behavior of this variable could disclose some interesting correlations between the ROM and passive torque at ROM at post (more than the absolute degree itself). If the Authors have this parameter, their are strongly invited to add these info as they could reinforce the study.

Other minors:

Line 159 and 221. Correlation with passive torque at DF ROM? From lines 147/148 it does not seem so....

Lines 171/172. Should not be the opposite, i.e., an increase in stretch tolerance contribues to an increase ROM?

Lines 203-208: This paragraph is quite unclear. Please rephrase.

Reviewer 3 Report

Relationship between stretching intensity and the effect of stretching on passive stiffness of gastrocnemius muscle-tendon unit

This study examined the relationship between relative or absolute stretch intensity and changes in the range of motion and passive stiffness after static stretching. Static stretching (120 s) was performed in sixty five volunteers. Range of motion before stretching was defined as 100% and the ratio of the angle during stretching was defined as the relative intensity. A significant correlation was found between absolute intensity and passive torque and also between relative intensity and range of motion and passive torque at dorsiflexion range of motion. It was assumed that a higher intensity might be effective in changing the passive properties of the muscle tendon unit. In contrast, a relative intensity might be effective in increasing range of motion, contributed by passive torque at dorsiflexion range of motion

GENERAL COMMENTS

The main concept of the study and the research purpose are clear. However, there are several issues that should be considered to bring this paper to publishable standards.

SPECIFIC COMMENTS

TITLE

  1. Please, consider rephrasing the title so as not to confuse the reader

ABSTRACT

  1. Line 25: …”stretching of the plantar flexors”. Please, correct.
  2. Line 26: …”for 120-s”. Please, correct.
  3. Line 27: please clarify ‘the given angle’.
  4. Line 28: range of motion at rest or during dorsiflexion? Please, clarify

INTRODUCTION

  1. Line 40: Please, rephrase to improve clarity
  2. Lines 68-71: Please, rephrase to improve clarity
  3. Lines 59-79: the authors are repeating the findings from the same studies. Please, provide a better rationale for the study.

METHODS

  1. Lines 89-93: Please, rephrase to improve clarity.

In addition, please clarify the ‘given angle’ for each participant. Just providing a reference is confusing the readers.

A schematic representation of the study protocol would also be beneficial.

  1. Line 99: what was 0° angle? 90° angle? ‘Neutral angle’?
  2. Lines 99 and 113: was the angle that participants felt no pain or discomfort, the maximum angle?
  3. Lines 116-118: Please, provide a schematic representation of the study protocol.
  4. Line 129: Please, provide exact p values.
  5. How was the number of participants calculated? E.g Power analysis?
  6. How did the authors calculate stretch intensity?
  7. Please provide reliability values (e.g ICC) for the tested variables.
  8. Please, also calculate effect sizes (e.g Cohen’s d) for pairwise comparisons
  9. It is known that sex (male or female) may affect range of motion. Why the authors have not performed separate analyses for males and females?

DISCUSSION

  1. Lines 159-162: In the Introduction section the authors mention several studies that examined different stretching intensities between participants. Please, correct.
  2. Lines 209-218: Please, rephrase to improve clarity and provide a reason for not performing separate analyses.
  3. A language revision throughout the manuscript would be beneficial

Round 2

Reviewer 1 Report

Thank you for your work on revising this manuscript.

Author Response

We thank the reviewer for the constructive comments and suggestions on our manuscript entitled “The relationship between stretching intensity and changes in passive properties of gastrocnemius muscle–tendon unit after static stretching.” Thanks to your comment, the paper has been improved.

Reviewer 2 Report

Authors replied adequately to all my previous issues. No further revisions are required.

Author Response

(The authors gave the same response as above.)

Reviewer 3 Report

I have no further comments

Author Response

(The authors gave the same response as above.)
